# Activity- and Enrichment-Based Metaproteomics Insights into Active Urease from the Rumen Microbiota of Cattle

**DOI:** 10.3390/ijms23020817

**Published:** 2022-01-13

**Authors:** Xiaoyin Zhang, Zhanbo Xiong, Ming Li, Nan Zheng, Shengguo Zhao, Jiaqi Wang

**Affiliations:** State Key Laboratory of Animal Nutrition, Institute of Animal Sciences, Chinese Academy of Agricultural Sciences, Beijing 100193, China; zhangxiaoyin1210@webmail.hzau.edu.cn (X.Z.); 82101201506@caas.cn (Z.X.); liming01@caas.cn (M.L.); zhengnan@caas.cn (N.Z.)

**Keywords:** urease, activity, metaproteomics, rumen

## Abstract

Regulation of microbial urease activity plays a crucial role in improving the utilization efficiency of urea and reducing nitrogen emissions to the environment for ruminant animals. Dealing with the diversity of microbial urease and identifying highly active urease as the target is the key for future regulation. However, the identification of active urease in the rumen is currently limited due to large numbers of uncultured microorganisms. In the present study, we describe an activity- and enrichment-based metaproteomic analysis as an approach for the discovery of highly active urease from the rumen microbiota of cattle. We conducted an optimization method of protein extraction and purification to obtain higher urease activity protein. Cryomilling was the best choice among the six applied protein extraction methods (ultrasonication, bead beating, cryomilling, high-pressure press, freeze-thawing, and protein extraction kit) for obtaining protein with high urease activity. The extracted protein by cryomilling was further enriched through gel filtration chromatography to obtain the fraction with the highest urease activity. Then, by using SDS-PAGE, the gel band including urease was excised and analyzed using LC-MS/MS, searching against a metagenome-derived protein database. Finally, we identified six microbial active ureases from 2225 rumen proteins, and the identified ureases were homologous to those of *Fibrobacter* and *Treponema*. Moreover, by comparing the 3D protein structures of the identified ureases and known ureases, we found that the residues in the β-turn of flap regions were nonconserved, which might be crucial in influencing the flexibility of flap regions and urease activity. In conclusion, the active urease from rumen microbes was identified by the approach of activity- and enrichment-based metaproteomics, which provides the target for designing a novel efficient urease inhibitor to regulate rumen microbial urease activity.

## 1. Introduction

Urease (urea amidohydrolase, EC 3.5.1.5), a nickel-dependent metalloenzyme, catalyzes the hydrolysis of urea into ammonia and carbon dioxide [1]. In ruminants, urea is commonly used as a cost-efficient replacement for feed proteins to provide the sole nitrogen source for urease [2,3]. Ammonia is converted by rumen microorganisms into microbial protein, which is a kind of superior protein, representing 60–85% of the protein in the small intestine [4]. However, the high catalytic activity of microbial urease contributes to the fact that the rate of urea hydrolysis is fourfold more rapid than that of ammonia synthesis [5], which not only reduces the utilization efficiency of urea, but also increases nitrogen emissions to the environment. Ruminant nitrogen emissions contribute to approximately 71% of the total nitrogen emissions of livestock [6]. In addition, high excreted nitrogen leads to drinking water pollution, which severely affects human and animal health [7]. Therefore, the key to making full use of urea is to make ammonia synthesize microbial protein as much as possible by regulating rumen microbial urease activity.

To accurately regulate urease activity, the discovery and identification of rumen microbial urease, especially the dominant urease, is particularly important. A conceptually straightforward approach to determine the prevailing enzymes is the isolation and selection of targeted enzymes from related environment samples using the culture-dependent method [8]. Some ureolytic bacteria with urease activity were isolated from the rumen [9], but the culture-dependent method limited the number of possible urease candidates. It was estimated that more than 99% of microorganisms were overlooked in the case of only pure culture [10]. The development of metaomics technology overcomes this problem. Using 16S rRNA gene sequencing of rumen bacteria and PCR amplification sequencing of urease genes, Jin et al. [11,12] showed the diversity of the rumen ureolytic bacterial community and found that a large number of urease genes were affiliated with *Paenibacillaceae*, *Helicobacteraceae*, *Clostridiaceae*, *Methylococcaceae*, and *Methylophilaceae* families. Through the use of metaproteomics, Stewart et al. [13] presented the largest dataset of predicted proteins from the rumen and provided an essential part for rumen microbiome structure and function. Hart et al. [14] indicated niche compartmentalization and functional dominance between abundant bacteria from the rumen. The method of metaomics contributes to the understanding and discovery of the rumen microbiota and rumen ureolytic bacteria, but the deficiency of the method is obvious, i.e., nontargeting. Although metaomics in theory was described as the entire genetic potential or the entire protein complement of the microbiota in the environment [15,16], it was eventually limited in use for the identification of targeted enzymes.

Enzyme activity plays a key role in the enzymatic reaction, but the identification of targeted enzymes with activity from complex samples remains challenging. To screen lipolytic biocatalysts with desired activity from soil samples contaminated with used cooking oil, Sukul et al. [17] proposed a simple workflow by using functional metaproteomics, in which they selected the active protein spots from gels separated by 2D-gel zymogram, and then lipolytic enzymes were identified by tryptic in-gel digestion and mass spectrometry of the active protein spots. Speda et al. [18] selected targeted active enzymes for bioprospecting from mixed microbial communities through 2D differential gel electrophoresis, and the difference from the identification method of lipolytic enzymes was that the differential active protein spots between an induced sample and a noninduced sample were identified using metaproteomics technology. Considering the complexity of the 2D-gel electrophoresis operation, Beller et al. [19] separated proteins using fast protein liquid chromatography (FPLC) instead of 2D-gel electrophoresis and then identified phenylacetate decarboxylase by comparative proteomic profiles of active FPLC fractions and those of adjacent inactive (or much less active) fractions. These studies on the identification of active targeted enzymes from different environments provide a potential appropriate methodology for the discovery of active urease in the rumen.

Given the diversity of microbial urease in the rumen, the discovery of dominant urease with high activity will be beneficial to accurately regulate rumen microbial urease activity and improve the utilization efficiency of urea. In the present study, extraction protocols of active proteins from the rumen microbiota were first evaluated to obtain the higher urease activity protein, and then the extracted protein was enriched by purification. The purified fraction with the highest urease activity was further separated by SDS-PAGE, and the gel band including urease was investigated by metaproteomic mass spectrometry-based tools. In addition, to improve the accuracy of urease identification, a metagenome-derived database of rumen microbiota was constructed. The discovery process of rumen microbial urease with high urease activity provides a potential method to identify targeted active enzymes from a complex environmental sample.

## 2. Results

### 2.1. Evaluating Extraction Protocols of Active Proteins from Rumen Microbiota

There are many protein extraction methods in metaproteomics studies. However, we focused on proteins with urease activity in the present study. Therefore, mechanical disruption methods were used for active protein extraction. To select the optimal performance of protein extraction methods, we set different parameters in each protocol of protein extraction. In all the protocols, ultrasonication, cryomilling, and high-pressure press generated more ammonia production (Figure 1B), among which ultrasonication and high-pressure press obtained the higher protein concentration (Figure 1A), but they obtained lower urease activity (Figure 1C). At the level of urease activity, the protocol of cryomilling showed the best performance, and the urease activity was the highest when cryomilling was carried out for 4 min. In addition, among the three pressures of the high-pressure press protocol, the 1000 MPa pressure yielded the highest protein concentration, ammonia production, and urease activity.

An ideal method of extracting active protein should not only produce protein with high urease activity, but also obtain increased protein extraction yield. The method of the highest urease activity (cryomilling for 4 min) was combined with that of the higher protein concentration and ammonia production (ultrasonication, high-pressure press for 1000 MPa); therefore, the following four combined methods were designed: 4 min + 112.5 W (sample was subjected to cryomilling for 4 min and then to ultrasonication using an amplitude of 112.5 W), 4 min + 137.5 W (as described above), 4 min + 162.5 W (as described above), and 4 min + 1000 MPa (sample was subjected to cryomilling for 4 min, and then treated using the high-pressure press with 1000 MPa). The higher protein concentration and ammonia production were obtained using the combined methods (Figure 1D,E), but the urease activity of the combined method was at least 40.37% lower than that of cryomilling for 4 min (Figure 1F). These results suggest that cryomilling for 4 min was a better protein extraction method for high urease activity than the other extraction methods, regardless of whether a combined method was used or not.

### 2.2. Enrichment of the Highly Active Urease

In order to obtain the protein with the higher urease activity, the ruminal microbial crude protein that was extracted by cryomilling (4 min) was separated using gel filtration chromatography. The purification chromatogram map is presented in Figure 2A, where a total of 12 fractions are illustrated in the *x*-axis. The two main peaks included Fractions 1–7 and 8–10, respectively. The protein distribution of the 12 fractions was analyzed using SDS-PAGE (Figure 2B and Appendix A). There were almost no protein bands in Fractions 1, 2, 11, or 12. A major band appeared in the 40–55 kDa of Fractions 4–10. Notably, from Fraction 7, a faint band appeared at 55–70 kDa, which was right in the molecular weight range of urease. The protein band of 55–70 kDa was clearer in Fractions 8–10, suggesting that urease may be concentrated in Fractions 8–10.

The urease activity of these 12 fractions was further determined. All fractions showed urease activity (Figure 2C), and the higher urease activity was presented in Fractions 8–10, which confirmed the hypothesis that urease was concentrated in Fractions 8–10. Therefore, we have selected Fraction 9 with the highest enzyme activity for metaproteomics analysis, in which only the single major band between 55 and 70 kDa from Fraction 9 was isolated to minimize the effects of miscellaneous protein.

### 2.3. Identification of Rumen Microbial Active Urease

An ideal protein database should contain the entire protein complement that might be expressed in the sample. In the present study, a metagenome-derived protein database of rumen microbiota was constructed, in which a total of 52.15 ± 6.71 Gb raw bases and 2,272,147 nonredundant protein sequences were generated. Fraction 9 from five dairy cows was subjected to metaproteomics analysis against the customized protein database. In total, 6905 peptides, 2225 proteins, and 1493 protein groups were identified. Among them, the core peptides and proteins were 927 and 799, comprising 29.04% and 57.36% of the corresponding mean, respectively (Appendix A), indicating good repeatability among samples.

Even with the identification of only the target band in the fraction with the highest urease activity, the protein composition was still complex. Six of the 2225 proteins were identified as urease (Figure 3A), and urease Lc1 was presented in sample1, sample2, and sample3 with the highest values of −10 lgP and coverage. According to a BLASTp analysis, Lc1 had the highest identity percentage of 94.1% to that of *Fibrobacter* sp. UWEL (NCBI ID: WP_073227198.1) (Figure 3B). Moreover, the result of the phylogenetic tree showed a distant phylogenetic affiliation between Lc1 and the known ureases, indicating that Lc1 may be an unclassified urease. The other five identified proteins (M1, M2, M3, M4, and M5) were classified as a protein group, which shared the highest identity percentage to that of *Treponema* sp. (NCBI ID: MBQ4013753.1).

### 2.4. Comparative Analysis of Identified Rumen Microbial Ureases and the Known Ureases

The active site and flap regions are closely linked to urease activity. The active site of urease is in the UreC and contains two nickel ions and a channel into the active site, which is protected by the flap. Flap regions comprised two α-helices and between them a flexible loop of β-turn. The amino acid sequences of ureases from rumen microbes, *Klebsiella Aerogenes* (*Ka*, PDB code: 4EP8), *Helicobacter pylori* (*Hp*, PDB code: 1E9Y), *Cajanus cajan* (*Cc*, PDB code: 4G7E), *Sporosarcina pasteur* (*Sp*, PDB code: 4CEX), and *Yersinia enterocolitica* (*Ye*, PDB code: 4Z42) were first aligned (Figure 4A). These residues at and near the active site of urease were well conserved, especially in the identified rumen microbial ureases (Lc1, M1, M2, M3, M4, and M5). For flap regions of urease, these residues were strictly conserved in the two α-helices, and the amino acid sequences of the α-helices in the ureases of Lc1, M1, M2, M3, M4, and M5 were identical, but these residues in the β-turn were not conserved. Considering that the ureases of M1, M2, M3, M4, and M5 were classified as a protein group, we selected one of these ureases, M3, to create a homology model. Among these known ureases (*Ka*, *Hp*, *Cc*, *Sp*, and *Ye*), those with high similarity to the β-turn sequences of Lc1 and M3 were selected for further structural comparison.

The 3D protein structures of Lc1, M3, *Ka*, and *Hp* are shown in Figure 4B. The ribbon diagrams of protein structures of Lc1 and M3 almost overlap (pink lines and rose red lines), and the position of the β-turn of Lc1 and M3 was evidently different from that of *Ka* and *Hp* (Figure 4C). Therefore, these residues at the active site and the two α-helices were conserved, and these residues in the β-turn were nonconserved, a finding that is of great importance and can be used to influence the flexibility of flap regions and urease activity.

## 3. Discussion

In the rumen, microbial urease plays a crucial role in urea metabolism, and proper regulation of urease activity contributes to the improvement of the utilization efficiency of urea and the reduction of nitrogen emissions to the environment. Dealing with the complexity of rumen microbiota and the diversity of rumen microbial ureases, it is very important to determine the dominant urease as the regulatory target. Enzyme activity is an important index to evaluate enzyme catalytic performance, especially in the present study, in which microbial urease with high urease activity may be an ideal regulatory target for improving urea utilization efficiency. Here, we presented an approach to discover microbial urease with high urease activity from the complex rumen through activity-based screening combined with metaproteomics identification.

In recent years, several studies on the activity-based screening of interesting enzymes have been reported. Sukul et al. [17] and Speda et al. [18] used 2D-gel electrophoresis to separate the samples; all protein spots were determined through in-gel activity assay, and the high-activity protein spots were selected for further identification using metaproteomics. In the first case [17], protein spots on 2D polyacrylamide gel electrophoresis needed to be refolded before activity could be detected. Speda et al. [18] used 2D differential gel electrophoresis, and each sample had to be labeled with a separate CyDye before loading onto a gel. Although all proteins were effectively isolated in the above studies, these complex operations and uncertainty of protein separation limited the application of 2D-gel electrophoresis. In the present study, by using a two-step screening method of protein extraction and protein purification, we generated the enriched protein fraction with the highest urease activity from the complex rumen.

Protein extraction in proteomics is an important step for ensuring reliable data [20]. In our study, the examined enzyme was screened based on its activity; so, we first set the urease activity as an indicator to assess different extraction protocols of active proteins from rumen microbiota. The protocol of cryomilling maximized urease activity with low protein concentration. Considering that the later step of protein purification would consume a large amount of protein, attempts to combine cryomilling with the protein extraction method with high protein concentration (ultrasonication and high-pressure press) were carried out. Unfortunately, the combined methods of protein extraction only increased protein concentration but did not improve urease activity. While these combinations did not result in the desired purpose, they did indicate that cryomilling was the better protein extraction method of high urease activity, and it could be a valuable choice to extract active protein from other environments. Additionally, it is also worth mentioning that ultrasonication and high-pressure press were two ideal methods for extracting protein with high protein concentration.

In fact, when we used cryomilling to extract protein, a large amount of protein was obtained through multiple extractions, which met the requirements of protein purification. Gel filtration chromatography was used to separate proteins of different molecular sizes. In the present study, these fractions with high urease activity were exactly those presenting a band of urease molecular size in the gel, indicating that urease could be enriched by protein purification of gel filtration chromatography. To discover a novel enzyme of catalyzing toluene biosynthesis, Beller et al. [19] and Zargar et al. [21] generated the fraction with the greatest toluene synthase activity using anaerobic FPLC purification, which was similar to the purpose of our purification. Considering that the protein composition of the desired fraction was complex, Beller et al. [19] carried out a comparative proteomic analysis of an active fraction and the adjacent inactive fraction through digestion of in-solution proteins. To minimize the effects of miscellaneous protein, we only excised the single major band including urease for metaproteomic analysis by digestion of in-gel proteins.

A suitable protein database is supposed to contain the entire protein complement that might be expressed in the sample [22], which plays a critical role in protein identification. Comprehensive sequence databases and a customized database through sequencing of the whole metagenome or metatranscriptome are two common strategies [23]. In the present study, we created a customized database including metagenomic sequences of rumen microbiota. Due to the complexity and diversity of microbial communities, it is very difficult to construct a suitable database. False discovery rate (FDR) was a standard for controlling the quality of protein identifications. An FDR of 1% was defined as the threshold [24], which improved the quality of accepting protein identifications and may reject true proteins [25]. To address this challenging issue, some workers put forward many optimization methods on the basis of the two kinds of databases [26,27]. Tang et al. [28] presented a graph-centric approach to improving metagenome-guided peptide and protein identification in metaproteomics, in which they used a two-step database search and reduced the excessive and potentially error-prone reference protein sequences. In the present study, we also used a two-step strategy for database search against the customized protein database and identified six ureases with a stringent FDR threshold. The one-step attempt to directly search against the customized database with the same control parameters was also performed, but no urease was identified.

As indicated, only six ureases were identified out of the 2225 proteins. The low urease identification rate may be related not only to the sample itself, but also to the protein database and the strict FDR threshold. The targeted band obtained through the two-step activity screening theoretically consisted of more active urease and other rumen microbial proteins of the same molecular size as the urease. However, there were only the metagenomics-based protein sequences, in which the abundance of urease was usually low. An increase in the sequencing depth and the targeting urease sequence could lead to the improvement of the identifications. Additionally, an FDR was constructed by using a target-decoy search method [29]. If there were many highly similar sequences in the protein database, the decoy database and the target database would not be able to separate these similar sequences accurately, resulting in difficulty in identifying the true and false results [30]. Urease sequences were highly conserved with higher similarity, an FDR of 1% may filter out some true ureases. Proteomics is a developing technology, and all strategies used for database construction and database search carried some advantages and disadvantages. Therefore, we should apply various approaches according to our own experimental purposes and sample conditions.

We identified two protein groups including six ureases with a 1% FDR threshold in the present study. One protein group was homologous to the known urease from *Treponema* sp. Three strains of *Treponema* sp. were firstly isolated from the cow rumen by Wozny et al. [31], and the three strains exhibited different urease activity in vitro. Another protein group had the highest identity percentage of 94.1% to that of *Fibrobacter* sp. UWEL. It has been found that the genus *Fibrobacter* plays an important role in cellulose decomposition and converting feed into volatile fatty acid [32,33]. The relationship between rumen microbial urease and *Fibrobacter* sp. has not been reported. The results of this study provided new insights into the function of *Fibrobacter* sp. in the rumen.

## 4. Materials and Methods

### 4.1. Collection of Ruminal Microbial Samples

Ruminal microbial samples from five dairy cows with ruminal cannulas were collected by filtration of rumen digesta fluid using four layers of cheesecloth. All collected samples from each dairy cow were quickly mixed and divided into 50 mL and 1.5 mL centrifuge tubes. They were then transported back to the laboratory using dry ice and stored in liquid nitrogen. The 50 mL aliquots were used for protein extraction, and the 1.5 mL aliquots were used for total DNA isolation. The animal study was reviewed and approved by the Animal Care and Use Committee for Livestock of the Institute of Animal Sciences, Chinese Academy of Agricultural Sciences (Permit Number: IAS2019-14).

### 4.2. Extraction of Active Proteins from Rumen Microbiota

The 50 mL aliquots of ruminal microbial samples were centrifuged at 300× *g*, 4 °C, for 10 min to remove protozoa and large particles. The supernatants were collected and again centrifuged at 12,000× *g*, 4 °C, for 5 min. The precipitated microbial cells were washed twice with ice-cold 50 mM 2-[4-(2-hydroxyethyl)piperazin-1-yl]ethanesulfonic acid (HEPES, pH 7.5) buffer. All washed microbial cells were then combined and resuspended in HEPES buffer, which was equally aliquoted and transferred to new centrifuge tubes for extraction of active proteins using different protocols as described below.

*Ultrasonication*: 10 mL of microbial cell suspensions were carried to ultrasonication for 10 min (6 s each with 6 s interval on ice) using an ultrasonic cell grinder (Scientz, Ningbo, China), in which three different amplitudes (112.5 W, 137.5 W, and 162.5 W) were set to select the optimum ultrasonication protocol of protein extraction.

*Bead Beating*: 200 mg zirconium beads (0.1–0.2 mm diameter) were added into the centrifuge tube containing 1.5 mL microbial cell suspensions. Beat beating was carried out using a TL2020 high-throughput tissue mill (DHS, Tianjin, China) at a speed of 1800 r/min for 4 min, 6 min, and 8 min (1 min each with 1 min interval at room temperature), respectively.

*Cryomilling*: 5 mL of microbial cell suspensions were subjected to cryomilling at a speed of 5 s (Retsch, Haan, Germany) in liquid nitrogen for 3 min, 4 min, and 5 min, respectively.

*High-Pressure Press*: 20 mL of microbial cell suspensions were crushed using the high-pressure press with 600 MPa, 800 MPa, and 1000 MPa, respectively.

*Freeze-Thawing*: The number of freeze-thawing cycles (3, 6, and 9) was assessed.

*Bugbuster Master Mix*: Microbial cells were lysed using Bugbuster Master Mix (Novagen, Madison, WI, USA) at 4 °C and at room temperature, according to the manufacturer’s instructions.

Samples treated by the above six protocols were centrifuged at 16,000× *g* at 4 °C for 10 min, and the collected supernatant was called ruminal microbial crude protein. The concentration of protein was estimated according to the Bradford Protein Assay Kit manufacturer’s protocol (Takara, Dalian, China). The ammonia release of ruminal microbial crude protein was measured by a modified phenol/hypochlorite reaction method [34]. One unit of urease activity was defined as 1 nmol of ammonia released per min per mg of ruminal microbial crude protein.

### 4.3. Enrichment of Active Urease by Gel Filtration Chromatography

Ruminal microbial crude protein was further enriched to improve the purity of active urease by gel filtration chromatography using a Superdex 75 column (GE Healthcare, Boston, MA, USA). The column was washed using 5 volumes of buffer A, pH 7.0, containing 150 mM HEPES, 0.5 mM ethylene diamine tetraacetic acid (EDTA), and 150 mM NaCl, with a flow rate of 1 mL/min. Ruminal microbial crude protein was concentrated by a 3 kDa centrifugal filter device (Millipore, Billerica, MA, USA) to a final volume of 200 mL and was loaded onto the washed column at a flow rate of 1 mL/min. The protein was eluted using buffer A, 5 mL of elution volume (the first 10 fractions) was collected, and the elution volumes of the 11th and 12th fractions were 50 mL and 10 mL, respectively. The urease activities of a total of 12 fractions were determined as described above. The protein distribution of the 12 fractions was shown using 12% SDS-PAGE.

### 4.4. LC-MS/MS Analysis

Metaproteomic analysis of the selected fraction with the highest urease activity was performed using trypsin digestion of the in-gel protein method. As described in detail [35], 55–70 kDa gel lane was decolorized using ammonium bicarbonate and acetonitrile and then reduced and alkylated using dithiothreitol and iodoacetamide, respectively. The processed gel lane was digested overnight using trypsin at 37 °C. The digested peptide was desalted, and LC-MS/MS analysis was carried out with a Q Exactive HF-X instrument (Thermo Fisher Scientific, Schwerte, Germany).

### 4.5. Metaproteomic Database Search and Urease Identification

Total ruminal microbial DNA was isolated using bead beating combined with the cetyltrimethylammonium bromide (CTAB) method [36]. The DNA library was constructed using the Nextera DNA Library Preparation Kit (Illumina, San Diego, CA, USA) and was used for sequencing on the Illumina Nova-PE150 platform of Qingdao Huada Gene Research Institute. Raw reads were preprocessed using Trimmomatic software to remove the adapters and low-quality reads (below quality 20 and length 50 bp) [37]. Reads with high quality were assembled into contigs using Megahit software [38]. The genes and proteins were predicted by Prokka software, and the nonredundant protein database was constructed by CD-Hit with a cut-off of 0.99 [39].

Peptide and protein identification were analyzed by PEAKS studio X plus software (Bioinformatics Solutions Inc., Waterloo, ON, Canada), with the precursor mass tolerance of 20.0 ppm and fragment mass tolerance of 0.05 Da. The maximum missed cleavages per peptide was set as two. The customized database used for the search was constructed using a metagenome-derived nonredundant protein sequence. To improve the peptide and protein identification in metaproteomics, the mass spectrometry raw data were subjected to a two-step database search. The first-step search was performed against the customized metagenome-derived protein database to generate a smaller refined database, in which the related parameters were set to 0. The reduced database was used for the second-step database search, with the score (−10 lgP) of peptide and protein ≥20, unique peptide ≥1, and a false discovery rate (FDR) of 1%.

### 4.6. Construction of Phylogenetic Trees for Identified Rumen Microbial Ureases

To construct the phylogenetic trees of identified rumen microbial ureases, we selected urease sequences that were most closely related to the identified rumen microbial ureases from the NCBI nonredundant protein sequence (nr) database by using BLASTp alignments, as well as the most common urease sequences from *Klebsiella Aerogenes* (*Ka*, PDB code: 4EP8) and *Helicobacter pylori* (*Hp*, PDB code: 1E9Y)) from the PDB database. The selected ureases and identified rumen microbial ureases were used to construct a phylogenetic tree using MUSCLE alignment [40] and the Neighbor-Joining method [41] by MEGA-X software [42].

### 4.7. Homology Modeling

The 3D protein structures of the identified rumen microbial ureases were created by homology modeling in the SWISSMODEL server [43]. For identified urease Lc1, urease from Jack bean (*Canavalia ensiformis*) (PDB code: 4GY7) shared 65.40% sequence identity with Lc1 and was used as the template to create a 3D molecular model of Lc1 with the global model quality estimate (GMQE) value of 0.81 and QMEANDisCo score of 0.75 ± 0.05. For the identified ureases of M1, M2, M3, M4, and M5, they had the same template from Jack bean (*Canavalia ensiformis*) (PDB code: 4GY7). The template shared 62.17–62.52% sequence identity with ureases of M1, M2, M3, M4, and M5, and the values of GMQE and QMEANDisCo used to evaluate model quality were 0.85–0.87 and 0.80–0.83, respectively. The 3D protein structures of identified rumen microbial ureases, and ureases from *Klebsiella Aerogenes* (*Ka*, PDB code: 4EP8) and *Helicobacter pylori* (*Hp*, PDB code: 1E9Y) were aligned using PyMol molecular graphics system.

## 5. Conclusions

In the present study, we present a simple method of determining microbial urease by targeting activity-based screening and enrichment from cattle rumen microbiota. We used a two-step activity screening approach to generate the ideal fraction with the highest urease activity, in which cryomilling combined with protein purification contributed to the extraction of rumen protein with higher urease activity. To minimize the effects of miscellaneous protein, the fraction with the highest urease activity was separated by SDS-PAGE, and only the gel band including urease was investigated using mass spectrometry. Six ureases were identified through a two-step database search against the metagenome-derived protein database. Furthermore, the 3D protein structures of identified ureases were shown using homology modeling, and as indicated, the β-turn of flap regions was very important to the flexibility of flap regions and urease activity.

The limitations of the study pertain to the difficulties in the construction of a suitable protein database. In a future study, we could try to improve the accuracy of identification by improving the depth of metagenomic sequencing or adding the number of targeted enzyme sequences through PCR amplification, probe capture, and other methods in the metagenome-derived protein database. In conclusion, the discovery process of rumen microbial active urease through interesting activity-based screening and enrichment expands the function of *Fibrobacter* sp. in the rumen and provides a potential method to identify targeted active enzymes from a complex environmental sample.

## Figures and Tables

**Figure 1 ijms-23-00817-f001:**
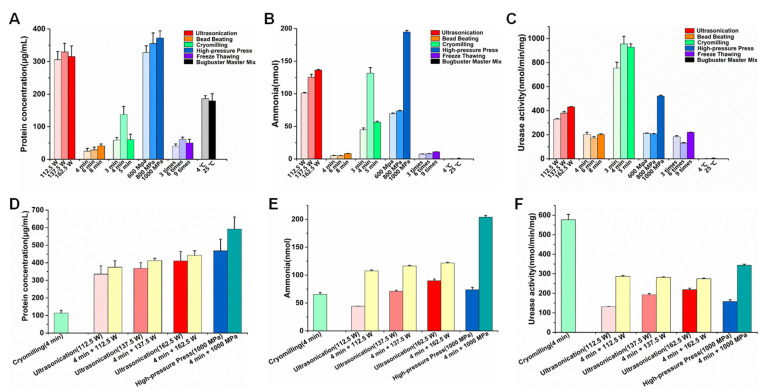
Optimization of extraction protocols of active proteins from rumen microorganisms. (**A**) Effects of different protein extraction methods on protein concentration. Different parameters were set in each protocol of protein extraction. Ultrasonication: three different amplitudes of 112.5 W, 137.5 W, and 162.5 W. Bead beating: three different durations of 4 min, 6 min, and 8 min. Cryomilling: three different durations of 3 min, 4 min, and 5 min. High-pressure press: three different pressures of 600 MPa, 800 MPa, and 1000 MPa. Freeze-thawing: three different cycles of 3 times, 6 times, and 9 times. Bugbuster Master Mix: two different temperatures of 4 °C and 25 °C. (**B**) Effects of different protein extraction methods on ammonia release. (**C**) Effects of different protein extraction methods on urease activity. (**D**) Effects of different combined methods of protein extraction on protein concentration. Cryomilling (4 min): Protein was extracted using cryomilling protocol for 4 min. 4 min + 112.5 W: Sample was subjected to cryomilling for 4 min, and then to ultrasonication using an amplitude of 112.5 W. Other methods were similar to those described above. (**E**) Effects of different combined methods of protein extraction on ammonia release. (**F**) Effects of different combined methods of protein extraction on urease activity.

**Figure 2 ijms-23-00817-f002:**
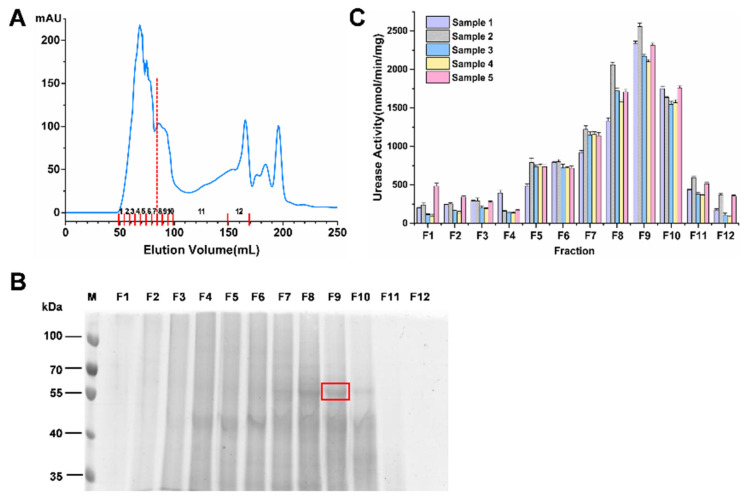
Purification of proteins with high urease activity. (**A**) Gel filtration chromatography map of proteins with high urease activity. The distance between the red lines represents the elution volume of each fraction. (**B**) SDS-PAGE gel of purified 12 fractions (F1–F12) from Sample 5. Protein lanes in the red box were isolated for metaproteomic analysis. (**C**) Urease activities of purified 12 fractions (F1–F12) from all samples.

**Figure 3 ijms-23-00817-f003:**
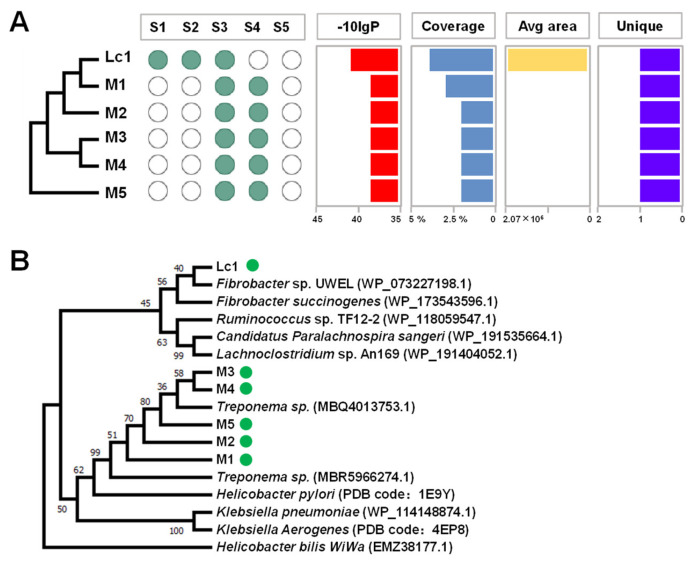
Discovery of rumen microbial urease by targeting activity-based screening. (**A**) Metaproteomic analysis of Fraction 9 with the highest urease activity. S1–S5 represents sample1–sample5. Different ureases were discovered in different samples (green dot) with different scores. −10 lgP: the PEAKS protein score. Coverage: the percentage of the peptide mapped to the identified protein. Avg area: average abundance of identified proteins. A zero in Avg area column means there is no feature detected but there is identification. Unique: unique peptide. (**B**) Phylogenetic trees of identified rumen microbial ureases. Numbers at nodes represent bootstrap support values. The accession numbers of protein in the NCBI GenBank database are in parentheses.

**Figure 4 ijms-23-00817-f004:**
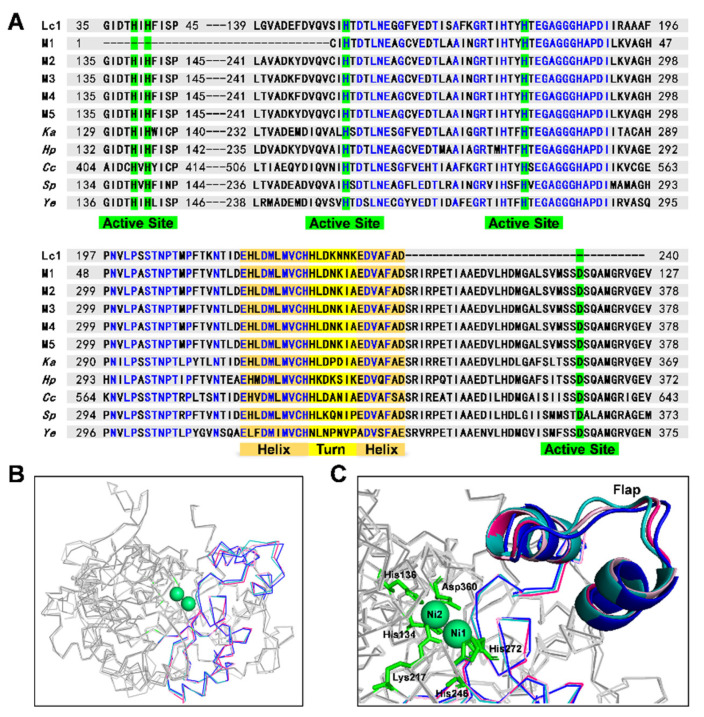
Alignment of identified rumen microbial ureases and known ureases. (**A**) Sequence alignment of identified rumen microbial ureases and ureases from *Klebsiella Aerogenes* (*Ka*, PDB code: 4EP8), *Helicobacter pylori* (*Hp*, PDB code: 1E9Y), *Cajanus cajan* (*Cc*, PDB code: 4G7E), *Sporosarcina pasteur* (S*p*, PDB code: 4CEX), and *Yersinia enterocolitica* (*Ye*, PDB code: 4Z42). Number represents the location of residue in their urease sequences. Residues identical in these all ureases are in blue. The flap (helix-turn-helix) regions are highlighted in orange (helix) and yellow (turn). The residues of active sites are highlighted in green. (**B**) Stereodiagram of identified rumen microbial ureases of Lc1 and M3 and ureases from *Klebsiella Aerogenes* and *Helicobacter pylori*. Their flap and flanking regions are colored in pink, rose red, cyan, and blue, respectively. Nickel ions are indicated as green balls. (**C**) Stereodiagram of active sites (in green) and flap regions from identified rumen microbial ureases of Lc1 (in pink) and M3 (in rose red) and ureases from *Klebsiella Aerogenes* (in cyan) and *Helicobacter pylori* (in blue).

## Data Availability

Not applicable.

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
