# Peer review of "Activity- and Enrichment-Based Metaproteomics Insights into Active Urease from the Rumen Microbiota of Cattle"

_ijms, 2022, doi:10.3390/ijms23020817_

Round 1

Reviewer 1 Report

The manuscript of Zhang et al. describes isolation and enrichment of a protein fraction with urease activity from cattle rumen. Fractions with highest activity were subjected to proteome analysis to identify enzyme candidates that may be responsible for the observed urease activity. The manuscript is an exhaustive description and discussion of the purification procedure. Identification and computational analysis of the enzyme candidates complement the study.

I consider this study as too preliminary for publication. Most importantly, the key experiments to demonstrate and thus verify urease activity of the retrieved candidates are missing. What remains to be done is heterologous expression, purification, and enzymatic analysis of the candidates. Because with the current data, there is no proof that the identified candidates are responsible for the observed activity in the protein mixture. In fact, there may be other ureases present that are responsible for the main activity, but where not detected with proteomics. Lastly, it may be that the identified urease candidates have no enzymatic activity. Therefore, proof of activity in classical expression/enzymatic assays is essential.

Furthermore, the English should be improved – the manuscript is understandable, but full of mistakes that can be eliminated by a native speaker or an editing service.

Some other points for consideration:

Fig 1 and text: You are showing the specific activity. But this must be set in relation to total activity. Because some of your extraction methods may yield much protein, but little activity and vice versa. It is best and classical approach for enzyme enrichment to show total activity per sample/fraction and specific activity.

Figure 3A: Avg Area is only shown for LC1, but not the other candidates. Explain/discuss.

Figure 4 and text: I recommend doing more sequence alignments to compare conserved and non-conserved regions. This can be easily done nowadays. Then you have stronger arguments for your hypothesis about flap regions and their suggested impact on activity.

Reviewer 2 Report

Manuscript ijms-1514253, entitled “Activity- and enrichment-based metaproteomics insights into active urease from the rumen microbiota of cattle”

Recommendation:       The above paper is not suitable for publication in its present form.

General Comments:

  • This article provides useful information about the application of activity- and enrichment-based metaproteomics for the determination of urease from the rumen microbiota of cattle. However, there are a lot of grammar, stylistic and syntax errors. In some cases, these errors negatively influence the understanding of the text. Please rephrase L31-32, 45-46, 160-161, 173-174, 406-410 etc
  • Please check reference style of the journal

Specific Comments:

L10: “Dealing with” instead of “Facing”

L11: “…urease and identifying…”

L13: What do you mean by “uncultured”? Not determined?

L13: “In the present study” instead of “Here” and “describe” instead of “present”

L15: “optimization method”

L16: “…among the six applied extraction…”

L18: “obtaining” instead of “extracting”

L23: “these” instead of “as”

L25: “…be crucial in influencing the flexibility…”

L34: “urea amidohydrolase”

L35: “In” instead of “Regarding the”

L37: “converted” instead of “synthetized”

L38: “representing 60-85%” instead of “according for 60%-85%”

L40: “contributes to the fact” instead of “resulting”

L44: “leads to” instead of “cases the”

L47: “To accurately regulate urease activity, the discovery…”

L49: “determining the prevailing” instead of “discovery interesting”

L53-54: “…overlooked in case of only pure…”

L54: “overcomes” instead of “counters”

L56: “showed” instead of “insighted into”

L59: “Through” instead of “By”

L60: “…rumen and provided…”

L63: “…to the understanding and discovery of the rumen…”

L64: “…non-targeting. Although, metaomics…”

L66: “…environment [15,16], it was eventually limited…”

L73: “…zymogram and then lipolytic…”

L80: “…electrophoresis and then identified…”

L87-89: “In the present study, extraction protocols of active proteins from rumen microbiota were first evaluated to obtain the higher urease activity protein and then the extracted protein was enriched…”

L98-99: “There are many protein extraction methods in metaproteomics studies. However, we focused on the protein with urease activity in the present study. Therefore…”

L109: “increased” instead of “more”

L138: “is presented” instead of “was depicted”

L139: “are illustrated in” instead of “were represented on”

L148: “hypothesis” instead of “inference”

L149: “have selected” instead of “choose”

L150-151: “which, only the single major band between 55-70 kDa from fraction 9 was isolated to minimize the effects of miscellaneous protein.”

L164: Please delete “respectively”

L165: “Among them” instead of “Of those”

L186: “are closely linked” instead of “were key”

L199: Please delete “were”

L202: “a finding that is of great importance and can be used” instead of “which might be key”

L218-219: “…contributes to the improvement of the utilization efficiency of urea and the reduction of the nitrogen emissions…”

L220: “determine the” instead of “discovery a”

L222: “…performance; especially in the present study, microbial…”

L224: “discover” instead of “discovery”

L226: “In recent years, several studies on…”

L227: “…reported. Sukul et al. [17] and Speda et al. [18] used 2D…”

L228: “…samples; all protein…”

L230: “the first case [17]” instead of “Sukul’s experiment”

L231: “Speda et al. [18] used…”

L235: “In the present study…”

L238: “is” instead of “was”

L239: “In our study, the examined enzyme…”

L246: “While these combinations did not result in the desired purpose, they did…”

L255: “In the present study…”

L258: “…Beller et al. [19] and Zargar et al. [21] generated…”

L261: “…Beller et al. [19] carried…”

L267-269: “sequencing of the whole metagenome or metatranscriptome are two common strategies [23]. In this study…”

L278: “In the present study…”

L282: “performed” instead of “done”

L283: “As indicated” instead of “In the study”

L283: “…of 2,225 proteins. The low…”

L288-290: “…was usually low. Increase of the sequencing depth and the targeting urease sequence could lead to the improvement of the identifications…”

L297: “apply” instead of “try”

L301: “…strains that were firstly isolated from the cow rumen by Wozny et al. [31], and the…”

L302: Please delete “[31]”

L321: “from each dairy cow”

L313: “They were then transported…”

L326: Please delete “Each”

L342: Please delete “respectively”

L345: “…min and the collected…”

L393-394: Please delete “not only”

L296: “the most” instead of “selected”

L405: Please delete “and”

L415: “In the present study, we present a simple method of determining microbial…”

L424: “…modeling, and as indicated, the β-turn…”

L427: “In a future study…”
